# Changes in the Dentate Gyrus Gene Expression Profile Induced by Levetiracetam Treatment in Rats with Mesial Temporal Lobe Epilepsy

**DOI:** 10.3390/ijms25031690

**Published:** 2024-01-30

**Authors:** Veronica Diaz-Villegas, Luz Adriana Pichardo-Macías, Sergio Juárez-Méndez, Iván Ignacio-Mejía, Noemí Cárdenas-Rodríguez, Marco Antonio Vargas-Hernández, Julieta Griselda Mendoza-Torreblanca, Sergio R. Zamudio

**Affiliations:** 1Departamento de Fisiología, Instituto Politécnico Nacional, Escuela Nacional de Ciencias Biológicas, Mexico City 07738, Mexico; vdiazv1000@alumno.ipn.mx (V.D.-V.); lpichardom@ipn.mx (L.A.P.-M.); 2Laboratorio de Neurociencias, Subdirección de Medicina Experimental, Instituto Nacional de Pediatría, Mexico City 04530, Mexico; noemicr2001@yahoo.com.mx; 3Laboratorio de Oncología Experimental, Instituto Nacional de Pediatría, Secretaría de Salud, Mexico City 04530, Mexico; ser.mend@gmail.com; 4Laboratorio de Medicina Traslacional, Escuela Militar de Graduados de Sanidad, Universidad del Ejército y Fuerza Aérea, Mexico City 11200, Mexico; ivanignacio402@gmail.com; 5Subdirección de Investigación, Escuela Militar de Graduados de Sanidad, Universidad del Ejército y Fuerza Aérea, Mexico City 11200, Mexico; mavh78@yahoo.com.mx

**Keywords:** epileptic rats, levetiracetam, microarray analysis, gene expression, calcium regulation

## Abstract

Temporal lobe epilepsy (TLE) is one of the most common forms of focal epilepsy. Levetiracetam (LEV) is an antiepileptic drug whose mechanism of action at the genetic level has not been fully described. Therefore, the aim of the present work was to evaluate the relevant gene expression changes in the dentate gyrus (DG) of LEV-treated rats with pilocarpine-induced TLE. Whole-transcriptome microarrays were used to obtain the differential genetic profiles of control (CTRL), epileptic (EPI), and EPI rats treated for one week with LEV (EPI + LEV). Quantitative RT–qPCR was used to evaluate the RNA levels of the genes of interest. According to the results of the EPI vs. CTRL analysis, 685 genes were differentially expressed, 355 of which were underexpressed and 330 of which were overexpressed. According to the analysis of the EPI + LEV vs. EPI groups, 675 genes were differentially expressed, 477 of which were downregulated and 198 of which were upregulated. A total of 94 genes whose expression was altered by epilepsy and modified by LEV were identified. The RT–qPCR confirmed that LEV treatment reversed the increased expression of *Hgf* mRNA and decreased the expression of the *Efcab1*, *Adam8*, *Slc24a1*, and *Serpinb1a* genes in the DG. These results indicate that LEV could be involved in nonclassical mechanisms involved in Ca^2+^ homeostasis and the regulation of the mTOR pathway through *Efcab1*, *Hgf*, *SLC24a1*, *Adam8*, and *Serpinb1a*, contributing to reduced hyperexcitability in TLE patients.

## 1. Introduction

Epilepsy is one of the most common chronic neurological disorders and affects more than 65 million people worldwide [1]. Temporal lobe epilepsy (TLE) is one of the most common focal types and is characterized by a state of neuronal hyperexcitability and spontaneous recurrent seizures (SRSs), with the dentate gyrus (DG) of the hippocampus being the key structure that acts as a “gateway” controlling the propagation of electrical discharges from the entorhinal cortex to areas such as CA3 [2,3,4]. Although the mechanisms underlying the occurrence of seizures in TLE patients are not fully understood, there is a classical theory that proposes an imbalance between the inhibitory and excitatory systems [5]. However, the mutations or changes in gene expression that have been associated with epilepsy do not directly alter the balance of inhibition and excitation, suggesting additional mechanisms involving fine molecular processes and the regulation of genes that continuously alter brain function and lead to ictogenic activity [6,7,8,9].

Patients with epilepsy are treated with antiepileptic drugs, which suppress seizures. Levetiracetam (LEV) is a second-generation antiepileptic drug that has demonstrated improved tolerability and improved efficacy; therefore, it has gradually become a first-line drug broadly used in clinics and is commonly used for treating TLE [10,11,12]. LEV is used alone or in combination with other medications for both focal onset and generalized onset seizure control [13]. In 1999, in the United States, the oral formulation of LEV was approved as an adjunctive therapy for the treatment of focal onset seizures, myoclonic seizures, and generalized onset [14]. In 2000, LEV was approved for the treatment of focal onset seizures and focal-to-bilateral tonic–clonic seizures in Europe as a single agent and as an add-on treatment for focal onset seizures, myoclonic seizures, and generalized onset tonic–clonic seizures [15].

The classical primary mechanism of action of LEV involves the modulation of synaptic vesicle glycoprotein 2A (SV2A), which is an integral membrane protein found in the vesicles of almost all synaptic terminals [16]. SV2A has been shown to function in both exocytosis and endocytosis processes in the synaptic vesicle cycle. During exocytosis, SV2A might function as a target for residual Ca^2+^ or influence the priming step, maintaining the availability of the secretory vesicles and ensuring correct neurotransmission [17]. During endocytosis, SV2A may regulate the adequate trafficking of the calcium sensor synaptotagmin protein and, consequently, Ca^2+^-stimulated fusion [18,19]. LEV may block the effect of SV2A by inhibiting its role in vesicular priming and thereby decreasing synaptic transmission [20]. Another possibility is that LEV acts to improve vesicular exocytosis, helping SV2A stabilize its conformation, which might optimize its general function [21,22,23]. Finally, LEV may enhance the role of SV2A in modulating the expression and trafficking of synaptotagmin [24,25].

Other targets may contribute to the effects of LEV; for example, it has been reported that LEV blocks voltage-dependent calcium channels, decreasing synaptic transmission [26,27,28,29]. In addition, LEV reduces potassium currents, inducing a decrease in repetitive action potential generation [30]. With respect to intracellular calcium systems, LEV reduces the calcium transients of the ryanodine and inositol triphosphate (IP3) receptors [31]. In the GABAergic system, LEV modulates region-dependent glutamic acid decarboxylase and increases GABA transaminase enzyme levels [32]. At the postsynaptic level, LEV blocks the effect of GABA_A_ receptor antagonists [33,34]. In the glutamatergic synapse, LEV modulates AMPA receptors and decreases the excitatory current [35]. Finally, LEV interacts with noradrenaline, adenosine, and serotonin receptors in the postsynapses involved in the pain system [36,37]. These effects involve different synaptic processes at the receptor and neurotransmitter levels with a unique profile not yet fully described, which together could contribute to the considerable efficacy of these treatments in partial and generalized epilepsies as monotherapy and/or polytherapy treatments [20,38].

Furthermore, LEV has shown antiepileptogenic properties and is thought to modify the expression of diverse genes in kindled rats [39,40]. For example, LEV treatment partially normalized the upregulation of genes related to synaptic plasticity and key modulators of seizure activity and neuronal reorganization, such as brain-derived neurotrophic factor, neuropeptide Y, glial fibrillary acidic protein, and thyrotropin-releasing hormone (*Trh*) [39,40,41]. In addition, the administration of LEV 1 h prior to kindling stimulation decreased the hippocampal overexpression of kindling-induced immediate early genes that encode transcription factors, neurotrophic factors, the transforming growth factor (TGF)-β superfamily, tumor necrosis factor (*Tnf*-α) and cyclooxygenase 2 (*Cox*-2) (two genes related to inflammatory processes) and proteins that regulate synaptic remodeling [39].

No studies have been carried out to determine the effect of LEV on the gene expression during chronic epilepsy. In addition, the lack of a change in the expression of the gene encoding the SV2A protein in TLE rats [8] indicates that new molecular targets could be involved in the effectiveness of LEV in the clinic. The objective of this study was to identify alterations in the transcriptional profile of the DG hippocampus of TLE rats via genome-wide array analysis as well as the effect of sub-chronic LEV administration on these changes.

## 2. Results

### 2.1. Effect of LEV on the SRS Number in Rats

Six weeks after the induction of status epilepticus (SE), the animals’ seizure behavior was evaluated for two weeks. Figure 1 shows the number of SRSs in each animal and the type of analysis that was performed on their DGs. At week seven (before treatment), the rats in the epileptic group (EPI) and the epileptic + LEV group (EPI + LEV) exhibited SRS. After that, during the treatment week (week eight), SRSs continued to be present in the EPI group; however, they were absent in the EPI + LEV group. It is important to note that, in the EPI group, the epileptic behavior during the two weeks of observation presented high variability between animals; the rats of this group showed an increase, decrease, or no change in SRS frequency (Figure 1). In particular, one rat in the EPI group did not exhibit SRSs during week 8; however, we considered all rats in this group to be epileptic based on previous reports [42] in which an animal is considered epileptic when at least one SRS is observed after the induction of SE (a chronic period of TLE), and such absence of seizures in this rat may be due to the high variability in the frequency of seizures among animals in this epilepsy model [42]. The previous consideration was supported by comparisons between the medians of the different groups. As is shown in Table 1, in the seventh week, the rats exhibited SRSs with medians of one seizure per week in the EPI group and two in the EPI + LEV group. During treatment (week eight), SRSs were absent in the EPI + LEV group; in contrast, the rats in the EPI group continued to present SRSs (medians 0 and 2, respectively, *p* < 0.05). Videos of the control (CTRL) rats during the observation period did not reveal any SRSs.

### 2.2. Gene Expression Profile in the Dentate Gyrus (General Features)

The quality of the microarrays was evaluated using principal component analysis (PCA). This allowed us to observe the general variability between groups. In Figure 2, clusters of microarrays with a covariance of 77.2% of the signal (CHP) data in the dataset were observed, indicating that there was a significant difference in the signal density of the microarrays with differential gene expression in the CTRL, EPI, and EPI + LEV groups.

In addition, using a heatmap, the differences in the gene expression profiles that changed according to epileptic condition were visualized. Similar changes in gene expression were observed among the EPI rats; however, some opposite areas were also found (Figure 3A). In the EPI + LEV group, there were changes in the expression of several genes that were modified by LEV treatment under epileptic conditions. In the heatmap of this group, several opposite green and red areas, with respect to those in the EPI conditions, were observed (Figure 3B).

Figure 4 shows the number of genes differentially expressed under epileptic conditions (EPI vs. CTRL). A total of 685 genes were modified (*p* < 0.05), 355 of which were downregulated and 330 of which were upregulated (Figure 4A,B). On the other hand, a total of 675 genes were differentially expressed after LEV treatment (EPI + LEV vs. EPI comparison), 477 of which were downregulated and 198 of which were upregulated (Figure 4A,B). A list of all the differentially expressed genes in the EPI vs. CTRL and EPI + LEV vs. EPI comparisons can be found in the Appendix A. The Venn diagram shows 94 shared genes (Figure 4A), that is, genes that were modified by epilepsy and altered by LEV treatment; these genes may be of special interest because they share both conditions.

As shown in Figure 5, the 685 genes (*p* < 0.05) differentially expressed under the EPI condition can be observed in the volcano plot, and a greater number of genes were found to have a range of fold change (FC) of 2 to 4 and −2 to −4. After applying a filter of FC ≥ 4, seven genes were found to be strongly overexpressed: importantly, complement component 3 (*C3*) (*p* = 0.000023), *Trh* (*p* = 0.05), S100 calcium-binding protein A11 (*S100a11*) (*p* = 0.0271), and tachykinin 3 (*Tac 3*) genes. The latter was the most highly expressed, with an FC of 12.49 (*p* = 0.022). Notably, the aforementioned genes have been previously reported in different types and models of TLE. On the other hand, for the underexpressed genes with an FC ≤ −4, 19 genes were observed, including solute carrier family 5 (choline transporter) member 7 (*Slc5a7*) and solute carrier family 4 member 11 (*Slc4a11*) (*p* = 0.0198), hemoglobin alpha 1 (*Hba1*) (*p* = 0.004), the olfactory receptor (*Olr*) 1691, and neuromedin B (*Nmb*) (*p* = 0.000017), which was the most underexpressed gene in TLE with an FC of −5.8.

In the second analysis, corresponding to the 675 genes modified by LEV treatment in epileptic rats (pink Venn diagram; Figure 4A), genes with FC values greater than 4 were also identified (Figure 6). To our knowledge, none of these genes, except for the complexin 4 (*Cplx4*) gene, have been linked to epilepsy. *Hba1* was the gene with the highest overexpression (FC 5.77: *p* < 0.0001), followed by *Cplx4* (FC 5.29; *p* < 0.0364) and *Olr1551* (FC 4.39; *p* < 0.0000283). Among the downregulated genes, novel genes previously not implicated in epilepsy, such as FERM domain-containing 7 (*Frmd7*) (FC −4.43 *p* < 0.000026), *Olr631* (FC −4.38 *p* < 0.0001), and tumor protein translationally controlled 1 (*Tpt1*), were found, with the last one being the most downregulated gene, with an FC of −6.6 (*p* < 0.0021) (Figure 6). Interestingly, the *SV2A* gene did not exhibit changes in expression even with less strict FC criteria.

### 2.3. Gene Ontology (GO) Analysis of the Epileptic Rats

According to the GO classification into three roots, (1) cellular component (CC); (2) molecular function (MF); and (3) biological process (BP), six clusters were found that met the selection criteria (see Section 4.6) and had biological importance to the pathology. In Figure 7A, the clusters are depicted, and the cluster with the highest enrichment score (E. score of 16.40) was related to integral components of the membrane, G-protein binding, and Olr activity and was associated with the highest number of modified genes (72 upregulated and 60 downregulated; 40 upregulated and 36 downregulated; 36 upregulated and 30 downregulated genes; Figure 7B,C). The second cluster, with an E. score of 2.66, was related to complement activation, positive regulation of apoptotic processes, and the inflammatory response; interestingly, we found that genes belonging to this cluster were mainly overexpressed (Figure 7D).

In the MF category, genes related to the potassium channel activity and cytokine activity were downregulated (Figure 7C); with respect to calcium ion binding, the number of genes with increased and decreased expression in this category was almost equal (5 upregulated and 6 downregulated). In the CC category, genes involved in glutamatergic synapses (Figure 7B,E) were found, some of which were prostaglandin-endoperoxide synthase 2 (*Ptgs2*), ATPase plasma membrane Ca^2+^ transporting 4 (*Atp2b4*), and calcium/calmodulin-dependent protein kinase IV (*Camk4*). On the other hand, genes associated with clusters related to the GABAergic synapses, synaptic vesicles and transport, and the postsynaptic membrane were mostly downregulated, especially genes located in the axon and its termination (Figure 7B). Considering that the same gene can belong to different GO processes and had been reported in experiments in two models of epilepsy (electrical kindling and kainic-induced SE), where alterations in these categories were described in the respective GO analyses [7,8,43], we performed a frequency analysis of the genes (Figure 7E); biological relevance, intersectionality, and uniqueness also were taken into consideration in the gene selection process. Interestingly, glycine receptor alpha 3 (*Glra3*) and dopamine receptor D1 (*Drd1*) were genes involved in more than two annotations related to epilepsy. Importantly, for the GABAergic synapses, gamma-aminobutyric acid type A receptor subunit rho3 (*Gabrr3*) and gamma-aminobutyric acid type A receptor subunit delta (*Gabrd*) were downregulated.

### 2.4. Gene Ontology (GO) Analysis of Epileptic Rats Treated with LEV

In the EPI + LEV rats, from significant GO annotations, terms were grouped into six clusters (Figure 8A): the cluster with the highest E. score (29.3) included G protein-coupled receptor activity, integral component of the membrane, and sensory perception of chemical stimuli. In this cluster, the highest number of genes were differentially expressed (*p* < 0.05), 50 upregulated and 36 downregulated; 76 upregulated and 47 downregulated; and 47 upregulated and 36 downregulated genes (Figure 8B,D). At the next level, the clusters included hemoglobin complex and oxygen transport with an E. score of 4.3, which were associated with both upregulated and downregulated genes (1 and 4; 2 and 3, respectively) (Figure 8A,C,D), followed by heme binding (E. score OF 3.3) with 5 upregulated and 5 downregulated genes (Figure 8A,B). Hemoglobin alpha, adult chain 1 (*Hba-a1*); hemoglobin subunit beta (*Hbb*); cytochrome P450, family 2, subfamily c, polypeptide 24 (*Cyp2c24*); cytochrome P450, family 4, subfamily a, polypeptide 2 (*Cyp4a2)*; hemoglobin subunit gamma 1 (*Hbg1*); cytochrome P450, family 4, subfamily a, polypeptide 8 (*Cyp4a8*); hemoglobin subunit epsilon 2 (*Hbe2*); and cytochrome P450, family 2, subfamily a, and polypeptide 2 (*Cyp2a2*) were genes that belonged to the oxygen transport category (Figure 8E).

Interestingly, these categories had not previously been associated with epilepsy or LEV treatment. In the following clusters, annotations pertaining to the positive regulation of the mitogen-activated protein kinase (MAPK) cascade (E. score 1.4), growth factor activity, and hormone activity (E. score of 0.9), in addition to non-clustered categories such as angiogenesis, showed higher frequencies of the following genes: leptin (*Lep*), fibroblast growth factor 8 (*Fgf8*), insulin-like growth factor 2 (*Igf2*), and hepatocyte growth factor (*Hgf*) (Figure 8A,E). Furthermore, for MF, CC, and BP, we observed that the genes associated with the GO terms growth factor activity, neuron projection, long-term synaptic potentiation, and negative regulation of gene silencing were only underexpressed (Figure 8B,D). It should be highlighted that this EPI + LEV vs. EPI analysis did not reveal genes related to complement activation or the GABAergic synapses, as would have been anticipated. However, it did show categories related to synapses and neuron projections, including genes such as myosin Vb (*Myo5b*), *Frmd7*, and solute carrier family 6 member 13 (*Slc6a13*). Finally, *Lep*, *Igf2*, and *Hgf* were the most representative genes that contributed to the highest number of enriched GO terms (Figure 8E).

### 2.5. Analysis of Genes Whose Expression Was Altered by Epilepsy and Modified by LEV Treatment

As shown in the Venn diagram (Figure 4A), 94 genes were affected by epilepsy and modified by LEV treatment. Among these 94 shared genes, the effect of LEV normalized the expression of 18 genes (Table 2): cystatin A (*Csta*); *Hgf*; Mas-related G protein-coupled receptor-X2 (*Mrgprx2*); serine (or cysteine) peptidase inhibitor, clade B, member 1a (*Serpinb1a*); reproductive homeobox 9 (*Rhox9*); zinc finger protein 53 (*Zfp53*); and *Olrs* (24, 260, 443, 576, 790, 1232, 1251, 1308, 1456, 1511, 1532, 1683). *Cplx4*, matrix metallopeptidase 9 (*Mmp9*), and the serpin peptidase inhibitor clade G (C1 inhibitor) member 1 (*SERPING1*) were overexpressed in both comparisons; however, LEV did not normalize the FC. The FC range in which most of the 94 genes oscillated was between −2 and −4 and +2 and +4. The 18 normalized LEV genes are associated with annotations of angiogenesis, inflammatory response, and Olrs.

To validate the microarray results, quantitative polymerase chain reaction (q-PCR) transcriptional analysis of eight genes—*Hgf*, *Serpinb1a*, EF-hand calcium-binding domain 1 (*Efcab1*), ADAM metallopeptidase domain 8 (*Adam8*), *Tpt1*, solute carrier family 6 member 13 (*Slc6a13*), solute carrier family 24, sodium/potassium/calcium exchanger, member 1 (*Slc24a1*), and Trh receptor (*Trhr*)—was performed. Since the objective of our study was to improve the understanding of the possible biological pathways transcriptionally associated with epilepsy and the response to LEV treatment, we used the following criteria to select candidate genes: (1) belonging to the group of genes shared by both comparisons (EPI vs. CTRL and EPI + LEV vs. EPI); (2) GO processes previously reported in epilepsy: we prioritized genes that exhibited significant changes in expression according to our microarray analyses and that also had known biological and functional relevance to epilepsy; (3) magnitude of differential expression: genes dramatically upregulated or downregulated and modified by LEV; (4) consistency with previous studies: we considered the changes in the expression of these genes against the findings reported in the scientific literature; and (5) diversity of gene functions: we included a variety of genes that spanned a variety of biological functions and processes. In some cases, the gene expression patterns were consistent between the qPCR and transcriptomic analyses; however, this was not observed for others (Figure 9, Figure 10 and Figure 11).

The *Hgf* and *Serpinb1a* genes were found to be differentially expressed in response to both factors via statistical analysis of the microarray data. Interestingly, the RT–qPCR analysis revealed that for *Hgf*, the relative expression of messenger ribonucleic acid (mRNA) increased in the EPI group compared with that in the CTRL group, and its expression normalized with LEV treatment (Figure 9A). No change in the *Serpinb1a* gene was observed between the EPI and CTRL groups; however, LEV treatment downregulated this gene (Figure 9B).

Among the genes involved in calcium ion binding (GO: 0005509), three genes were validated: LEV treatment reduced the mRNA expression of the *Efcab1* and *Adam8* genes in TLE rats (Figure 10A1,A2,B1,B2), and *Tpt1* gene expression was also reduced by LEV treatment, but no significant difference was observed (*p* = 0.1037) (Figure 10C1,C2).

Finally, another group of genes belonging to the solute carrier transporter family was analyzed; here, LEV reduced the relative expression of *SLC24a1* but not that of the *SLC6a13* or *Trhr* genes (Figure 11).

## 3. Discussion

According to our ontological analysis, genes related to complement activation, positive regulation of the apoptotic process, the inflammatory response, and calcium ion transmembrane transport were highly overexpressed, similar to previous findings in other TLE models (Figure 7D) [44,45,46]. Inflammation has been shown to occur in epilepsy, where neuroinflammation accompanied by neuronal loss and gliosis has been established as a significant element in the pathogenesis of seizures [47]. In TLE, neuronal death has pro-epileptogenic effects, as indicated by the identification of genes regulating apoptosis via signaling pathways such as JAK-STAT, phosphoinositide 3-kinase (PI3K), and mammalian target of rapamycin (mTOR), which are related to the G protein receptors involved in signal transduction [48]. Notably, integral components of the membrane and G protein-binding proteins, both of which are related to receptors, were highly expressed (Figure 7B,C). Furthermore, with inflammation, alterations in calcium-binding proteins and voltage-dependent calcium channels have been observed as part of the pathological development of epilepsy [49]. In relation to those in the EPI + LEV group, high G protein-coupled receptor activity, integral components of the membrane, and sensory perception of chemical stimuli were observed (Figure 8A). In the presence of LEV, complement activation, positive regulation of apoptosis, the inflammatory response, and calcium ion transmembrane transport (observed in the EPI group; Figure 7D) were decreased or not expressed, indicating that the mechanism of action of this drug targets these processes, probably through the regulation of receptors related to protein G [50].

On the other hand, some patients with TLE experience a series of signs and symptoms prior to the onset of seizures, which in clinical practice are called auras. In this sense, olfactory auras have been related to structural abnormalities such as reductions in gray matter, the amygdala, and the hippocampus in parallel with the activation of epileptogenic focus via afferent projections of the olfactory nuclei [51,52]. In this context, the altered *Olr* activity, which was highly expressed in the EPI group (Figure 7A,C), is consistent with the pathophysiological process of epilepsy. Notably, LEV normalized the expression of multiple genes related to *Olrs* (Table 2), which contributed to the normalization of *Olr* activity.

In relation to the analysis of the gene expression levels, studies of hippocampal samples from patients with mesial TLE with hippocampal sclerosis have shown changes in genes related to synapses, ion channels, and glutamate receptors [53,54]. In 2017, Dingledine et al. reported that 73 genes were commonly dysregulated in the acute phase in three different animal models of TLE (pilocarpine, kainic acid, and electrical kindling); our results matched 15 genes in the EPI group: CXADR-like membrane protein (*Clmp*), nescient helix-loop-helix 1 (*Nhlh1*), activin A receptor type 1C (*Acvr1c*), growth differentiation factor 10 (*Gdf10*), glypican 3 (*Gpc3*), sodium voltage-gated channel beta subunit 4 (*Scn4b*), neurotrophin 3 (*Ntf3*), polo-like kinase 5 (*Plk5*), *Gabrd*, 5-hydroxytryptamine (serotonin) receptor 5B (*Htr5b*), Nmb, *Mmp9*, core-binding factor beta (*Cbfb*), SH2 domain-containing adapter protein B (*Shb*), and *Trh* (Figure 7E), which might suggest that there are key genes involved in epilepsy since they are deregulated in both the acute and chronic phases of the disease [55]. Furthermore, we observed a decrease in the expression of genes related to the GABA_A_ receptor subunits, such as *Gabrr3* and *Gabrd*, in the epileptic rats (Figure 7E). Similarly, a microarray study in a tetanic stimulation model revealed the downregulation of the *Gabrd* gene in hippocampal formation during epilepsy [8]. Moreover, in humans, mutations in the *Gabrd* gene (variation c.659G > A; Arg220His and Glu177Ala) resulted in decreased GABA_A_ delta receptor current amplitudes, causing a predisposition to syndromic idiopathic generalized epilepsy and generalized epilepsy with febrile seizures [56,57]. In contrast, the LEV-treated rats did not show any change in the gene expression in the GABAergic system, although it has been reported that LEV increases the vesicular release of GABA through a nongenetic mechanism, binding to the SV2A protein [58]. There are no reports of LEV-mediated modulation of *Gabrr3* or *Gabrd*; however, it has been reported that LEV does not affect GABA-evoked currents in the hippocampus but does in the neocortex [59,60]

Regarding the inflammatory response, it has been widely reported that seizures induce a wave of inflammation in the brain; activate endothelial cells, glia, and astrocytes; and induce the upregulation of the inflammatory cytokines interleukin (IL)-1β, TNFα, IL6, complement components (C1q-C4), prostaglandin E2 (PGE2), and COX-2 [61]. Indeed, genes encoding these proteins have been found to be overexpressed in several preclinical and clinical studies of epilepsy, revealing evidence of aberrant inflammation, neurotoxicity, and hyperexcitability in the neuronal circuits. Moreover, these genes have been highlighted as possible diagnostic markers due to their high reproducibility [8,55,62,63]. The above findings agree with our results of observing clusters of complement system activation genes, increased inflammatory responses, and the overexpression of the *C4b*, *C4a*, *Ptgs2*, and *C3* genes in the EPI rats compared with those in the CTRL group (Figure 7A,D,E). Interestingly, LEV treatment attenuated the overexpression of several genes related to the inflammatory process (*SERPING1* and *Serpinb1a*). It has been suggested that LEV can attenuate the expression of genes by inhibiting epileptiform neuronal activity in the form of spikes in CA3 [64] and GD [65], which has been associated with changes in gene expression [66,67]. With respect to the SERPING1 components and Serpinb1a, no studies have examined the effect of LEV on the expression of these genes. However, recent evidence has suggested that LEV, which has been used in different animal models and phases of epilepsy, exerts neuroprotective effects via its anti-inflammatory effects [68,69,70]. Treatment with LEV suppresses the expression of pro-inflammatory molecules, such as TNFα, IL-1β, IL6, and inducible nitric oxide synthase, 1 or 3 h after SE [71], and epileptogenesis prevents microglial activation, as evaluated by a decrease in morphological changes, phagocytic activity, and cytokine expression [69]. Additionally, in epileptic rats, LEV reduces reactive gliosis and the expression levels of IL-1β and interleukin 1 receptor type I in the hippocampus and piriform cortex [68]. Finally, the anti-inflammatory effect of LEV was also demonstrated using in vitro assays in microglial/astrocyte co-cultures. LEV treatment restores astroglial gap junction coupling, along with repolarization of the membrane rest potential [72], and increases the expression of growth factor beta 1 [73]. Taken together, the above observations suggest that the efficacy of LEV is derived in part from its ability to prevent astroglial inflammatory activity. In contrast, LEV treatment did not reduce the complement components or proinflammatory cytokine expression, which might indicate that its mechanism of action does not classically reduce neuroinflammation once seizures are established. Additional investigations are necessary to address this issue.

New emerging studies have shown that the neuroinflammation caused by activated glial cells promotes neuronal hyperexcitability and seizures and is a crucial factor in the disruption of the blood–brain barrier (BBB). Moreover, pathological angiogenesis is also associated with the breakdown of the BBB, which has been found both in patients with intractable TLE and in epileptic rodents, and these alterations are prominent in the chronic phase and are likely triggered by recurrent seizures [74,75,76]. We observed that GO angiogenesis was significantly modified in both comparisons (EPI vs. CTRL and EPI + LEV vs. EPI; Figure 7D and Figure 8D), particularly in the EPI + LEV group, and several genes involved in angiogenesis, such as *Adam8*, were downregulated (Figure 10B1,B2), which suggests a certain protective effect of LEV against BBB disruption since recent evidence has shown that reducing the expression of *Adam8* (a gene related to metastasis by promoting proliferation, migration, and angiogenesis through αvβ3/PI3K/protein kinase B (AKT) integrin signaling in tumor cells) could promote vascular remodeling and improve the BBB integrity [71]. There is evidence that LEV administration can temporarily inhibit the BBB rupture caused by pilocarpine-induced SE through the downregulation of angiogenic factors such as angiopoietin 2/Tie2 receptor, vascular endothelial growth factor, and vascular endothelial growth factor receptor 2 [71]. This result is in line with previous observations that after SE, repeated high doses of LEV prevent the development of brain edema in the limbic regions. Although how LEV induces this protective effect is unclear, it has been proposed that the mechanism involves the protection of the integrity of the BBB [77]. Furthermore, in an animal model of neuroinflammation, administration of a disintegrin domain of Adam8 suppressed its progression, most likely through the inhibition of Adam8 in the central nervous system [78], suggesting that *LEV-mediated ADAM8* gene depletion has a beneficial effect on the inflammation caused by SRS.

Furthermore, the exact mechanism by which LEV reduces hyperexcitability and SRS generation remains unclear; we identified two novel genes, *HgF* and *Serpinb1a* (Figure 9), that could be related to the mechanism of LEV action, which has not been previously described. There is evidence that *HgF* and *Serpinb1a* can activate the mTOR pathway. The activation of this pathway has been implicated in SE models, in which a peak occurs at 3 h after SE onset [79]. Moreover, hyperactivation of the PI3k/AKT/mTOR pathway in different cell types (granular cells, glia, and hilar neurons) promotes neuronal hyperexcitability and SRS generation [79,80,81]. The HGF/MET receptor mediates the activation of several downstream signaling pathways, the PI3k/AKT/mTOR pathway being among the main ones [82]. Other studies have shown that the AMPK/mTOR pathway is activated by *Serpinb1a* overexpression in ischemia models and that upstream ADAM causes AKT activation [83,84]. In this regard, we observed an increase in the *Hgf* gene in the EPI group (Figure 9) and a decrease in *Hgf*, *Serpinb1a*, and *Adam8* gene expression in the LEV-treated group (Figure 9 and Figure 10B1,B2). We subsequently hypothesized that the reduced expression of these genes induced by LEV treatment reduces the hyperactivation of the PI3K/AKT/mTOR pathway, which might reduce the amount of apoptosis, neurogenesis, mossy fiber sprouting, and hyperexcitability in epilepsy (Figure 12).

Another widely discussed therapeutic target in TLE is calcium-mediated signaling. In rats with chronic epilepsy, the basal levels of intracellular calcium are elevated, and calcium levels are decreased in the extracellular space; together, these conditions lead to a decreased ability to modulate transient calcium fluxes and neurotoxicity, apoptosis, and seizure maintenance [85]. The regulatory effect of calcium homeostasis on ictal activity is even more complex because it interacts not only with neuronal activity but also with nonneuronal cells (astrocytes and glia) [86]. Several molecular components are responsible for maintaining intracellular calcium homeostasis; these include voltage-dependent calcium channels, ER-resident IP3 receptors, ryanodine receptors, and calcium-binding proteins and are involved in the calcium dysregulation underlying epileptic seizures [87,88]. In an animal model of TLE induced by kainic acid in mice, Cav3.1 T-type calcium channels were shown to play a modulatory role in the duration and frequency of hippocampal seizures, as well as epileptogenicity, mostly during acute periods [89]. This finding is consistent with our result that the calcium voltage-gated channel subunit alpha1 g (*Cacna1g*) gene encoding Cav3.1 was overexpressed in epileptic rats (Figure 7E). With respect to LEV treatment, contrary to expectations, since LEV is able to inhibit the N-, L-, and P/Q-type calcium channels, leading to a decrease in the intracellular Ca^2+^ concentration, our results did not indicate a change in the expression of genes encoding voltage-gated calcium [3,11,20,90]. Nevertheless, in the EPI + LEV group, we observed changes in the expression of genes associated with calcium-binding activity and growth factors such as *Tpt1*, *Efcab1*, protease, serine, 2 (*Prss2*), sushi domain containing 1 (*Susd1*), and *Hgf* (Figure 8E), which modulate calcium signaling. In particular, we validated a decrease in the expression of the *Hgf* and *Efcab1* genes (Figure 9A and Figure 10A1,A2). Exogenous Hgf treatment of the hippocampal neurons stimulates Ca^2+^ entry into the dendrites through the N-methyl-D-aspartate (NMDA) receptor and promotes the formation of IP3 and 1,2-diacylglycerol by stimulating phospholipase C, causing an increase in intracellular Ca^2+^ [91,92]. Thus, the decrease in *Hgf* might be another mechanism by which LEV reduces neuronal hyperexcitability (Figure 12). Additionally, the expression of *Efcab1*, a novel Ca^2+^-dependent neuronal sensor in rodents, was decreased in epileptic rats treated with LEV. Although the function of *Efcab1* in the hippocampus is not known, studies have shown that *Efcab1* is related mainly to ciliary dyskinesia and sperm chemotaxis [93,94]. Another study indicated that Efcab1 may have an anti-arrhythmic effect through CaV1.2 channel blockade [95]; LEV may also be involved in intracellular calcium regulation through *Efcab1* (Figure 12). Other authors mention that calcium-induced calcium homeostasis may be a key mechanism of LEV by reporting a 74% decrease in IP3 receptor-mediated calcium currents [31]. In addition, our results revealed a decrease in the *Slc24a1* gene in the EPI + LEV group (Figure 11B1,B2); this gene encodes the NCKX1 protein. The SLC24 family contains different K^+^-dependent Na^+^-Ca^2+^ exchangers that use the electrochemical gradients of both Na^+^ and K^+^ to drive the efflux of Ca^2+^ [96]. These exchangers are key determinants of Ca^2+^ regulation, especially in environments where the ion concentrations undergo large changes, such as excitatory cells and transport epithelia [97]. As a bidirectional Ca^2+^ exchanger, NCKX1 may play a relevant role in epilepsy since its inhibition might limit the NMDA-mediated excitotoxicity in cerebellar neuronal cultures [98].

We also found genes related to hormonal activity; metabolic alterations have been reported as a consequence of both epileptic conditions and drug treatment and are specifically associated with weight gain and the affectation of thyroid hormone levels, which not only have a psychological impact on patients but also vascular complications. Mainly in epileptic patients, dysregulation of the hormones leptin (Lep) and insulin has also been observed [99,100,101]. Lep markedly influences the excitatory synaptic transmission and synaptic plasticity in CA1 neurons [102]. In the EPI + LEV group, the *Lep* gene was related to various cellular processes according to our ontology analysis, and its expression was decreased compared with that in the EPI group (Figure 7E); however, this decrease was not statistically significant.

On the other hand, in the EPI group, we found the overexpression of another gene related to hormonal activity, the *Trh* gene (Figure 5 and Figure 7E), whose expression profile showed a biphasic effect in models of seizures induced by amygdalin kindling [102]. In addition, in areas such as CA3 and the entorhinal cortex, the *Trh* gene was overexpressed in a tetanic stimulation model of TLE [8,40], which agrees with our results observed in the GD. However, in a kindling model, intranasal administration of a Trh analog acutely suppressed seizures in a dose-dependent manner [103]. According to our differential expression profile induced by LEV treatment, we did not observe a difference in the expression of the *Trh* gene; however, *Trhr*, as well as *Lep*, tended to be underexpressed, and LEV treatment may have little effect on the metabolic outcomes of epileptic rats.

The downregulation of the *Nmb*, *Orl1691*, *Slc4a11*, and *Slc5a7* genes and the upregulation of the *C3*, *Tac3*, and *S100a11* genes observed in the EPI group (Figure 5) might induce the inactivation of GABAergic transmission; an increase in the frequency of epileptiform activity; olfactory dysfunction (impairment in odor discrimination and odor identification); an increase in neuronal injury and inflammation; activation of glial cells; and a lack of control of solute homeostasis, effects that characterize epilepsy [104,105,106,107,108,109]. Moreover, the presence of LEV decreased *Frmd7*, *Olr631*, and *Tpt1* gene expression and increased *Hba1*, *Olr1551*, and *Cplx4* gene expression (Figure 6), indicating that this drug could be involved in control over the organization of the membrane proteins of the cytoskeleton, neuronal growth and development, the regulation of cell proliferation, and the modulation of the release of neurotransmitters, and probably reestablished olfactory function, as previously mentioned [110,111,112], but did not improve the number of seizures [113].

Our findings indicate that LEV might modulate signal transduction processes related to the second messenger Ca^2+^-G-protein, inflammation, redox control, apoptosis, and glutamatergic neurotransmission. Moreover, LEV normalized the expression of genes related to immunomodulation, neuronal damage, gliosis, neuronal plasticity, molecular reorganization, neurogenesis (*Csta*), the modulation of GABAergic inhibition and seizure susceptibility (*Hgf*), neuropeptide ligands and immune response (*Mrgprx2*), the modulation of neurotrophic factors, cell death and inflammation (*Serpinb1a*), cell differentiation (*Rhox9*), the regulation of gene transcription and translation, neuronal development and synaptic activity (*Zpf53*), and olfactory tasks (*Olr*) (Table 2) [105,114,115,116,117,118,119]. The overexpression of the *Cplx4*, *Mmp9*, and *Serping1* genes after drug administration (Table 2) seems to indicate the role of LEV in neurotransmitter release, the remodeling of brain circuitry and synaptic plasticity, neuroinflammation, and cell death [120,121,122]. These findings may facilitate the identification of candidate genes for the treatment of TLE and provide data that can help us to better understand LEV’s mechanisms of action and identify new neuroprotective effects of this drug.

Finally, although the current results are promising and provide valuable information, it is important to highlight the limitations of this study due to the number of animals used for the microarrays (3–4 rats) and due to the variability in the data, given the proximity of *p* values to the significance threshold for some genes. A larger sample size could help minimize the variability and provide a more robust assessment of the genes of interest. Thus, we recommend that future studies with larger samples be conducted to validate and expand our findings. Nevertheless, it is important to consider that the observed variability may also reflect the biological heterogeneity inherent in animal models of epilepsy and in response to pharmacological treatments such as LEV. Despite these limitations, we believe that our findings significantly contribute to the understanding of how LEV modifies gene expression in TLE patients and provide a basis for further detailed investigations. Additionally, our findings could facilitate the identification of new candidate genes for the treatment of TLE and provide data that may help us to better understand the mechanisms of action of LEV and identify new neuroprotective effects of this drug.

## 4. Materials and Methods

### 4.1. Animals

Male Wistar rats (purchased from the Instituto de Investigaciones Biomédicas, Mexico City, Mexico) weighing 250–300 g at the time of SE induction were used. The animals were housed in acrylic boxes under the standard conditions of a controlled environmental temperature (22 ± 2 °C) and a 12 h light/dark cycle (lights on at 6:00 a.m.), with food and water available ad libitum. The rats were randomly assigned to the following groups: (a) CTRL, (b) EPI, and (c) EPI + LEV, as indicated in Table 3; the techniques performed on the brain of each rat (microarray analysis, qPCR analysis, or both) are also shown. All the experimental procedures were performed in accordance with the National Institutes of Health Guide for the Care and Use of Experimental Animals and Mexican law (SAGARPA NOM-062-Z00-1999). The protocol was approved by the Local Institutional Committees of the Instituto Nacional de Pediatría (INP; registered number INP 2020/003).

### 4.2. Induction of Status Epilepticus

To induce SE, the EPI and EPI + LEV rats were administered lithium chloride (127 mg/kg, i.p.; Sigma-Aldrich; Darmstadt, Germany) 19 h before pilocarpine to reduce the mortality [58,123]. To prevent any deleterious peripheral effects, the animals were injected with scopolamine methyl bromide (1 mg/kg, i.p.; Sigma-Aldrich) 30 min before the pilocarpine administration. Then, SE was induced using pilocarpine hydrochloride (30 mg/kg, i.p.; Sigma-Aldrich). The CTRL group only received 0.9% NaCl (PISA, Guadalajara, Mexico) at each time point. SE was defined as continuous seizures for more than 30 min, reaching stage 4 or 5 on the Racine scale at the beginning and remaining at least in stage 3 for 30 min [124]. After 90 min of SE, the rats were injected with diazepam (5 mg/kg, i.m.; PISA) to suppress their seizure activity and were placed on an ice bed for 1 h to reduce hyperthermia. Eight hours after the first dose, the animals received a second dose of diazepam (5 mg/kg, i.m.). Finally, the rats received 5 mL of 0.9% NaCl (s.c.) for rehydration and were housed overnight at 17 ± 2 °C. From the second day until the fifth day, they were fed Ensure^®^ food as part of their post-SE care and were monitored constantly until their recovery, when they entered a period of apparent well-being [123].

### 4.3. Behavioral Monitoring of Spontaneous Recurrent Seizures

Video monitoring of the SRSs began seven weeks after SE and was performed from 8:00 am to 6:00 pm. This interval was based on a prior study and our own observations of a higher SRS frequency during the diurnal period [42,58,125]. The video monitoring period lasted 2 weeks (1 week before LEV treatment and 1 week during treatment) and was performed using a DVR system with 4 cameras (Steren model CCTV-970; México City, México). The video analysis was performed using the H.264 playback program for Windows (v.1.0.1.15; Infinova, Guangdong, China). The seizure number was detected using fast-forward speed (8×), but seizures were verified at real-time speed.

SRSs were defined as behavioral seizures with Racine scores ≥ 3 (clonus of the anterior extremities, chewing or “wet dog shakes”, and clonic seizures with a loss of posture or uncontrolled jumps) [126]. Rats exhibiting SRSs during the week prior to LEV treatment were considered epileptic [124].

### 4.4. Levetiracetam Treatment

Seven weeks after SE, osmotic mini-pumps (ALZET^®^, 2ML1; Cupertino, CA, USA; release rate 10 μL/h) were implanted subcutaneously in the EPI + LEV group to provide sub-chronic LEV treatment for one week (300 mg/kg/day) [58,123,127]. It should be noted that a CTRL + LEV group was also included in this studysince our interest was focused on describing the changes in LEV administration in the epileptic brain, and there were no significant differences in the genes validated in this group compared to those in the CTRL group. Then, the rats were anesthetized with 5% isoflurane (with a 2 mL/min O_2_ flow) for induction and 2–3% for maintenance. A small skin incision was made at the level of the left scapula to implant the minipump under the skin. To fill the mini-pump chamber, the LEV was extracted from tablets (UCB Laboratories; Brussels, Belgium). Two tablets (1000 mg) were dissolved in 3 mL of 0.9% NaCl (PISA, Mexico City, Mexico), and the solution was sonicated for 10 min and centrifuged at 3000 rpm for 15 min (Hermle Z 326, Labnet, rotor 220.72). The supernatant was filtered (0.45 μm; Corning, NY, USA), and the mini-pumps were filled according to the manufacturer’s instructions. After implantation, the incisions were sutured and cleaned. The animals were administered a single dose of antibiotics (enrofloxacin; 5 mg/kg, i.m., Baytril^®^, Bayer, Leverkusen, Germany), analgesics (meglumine flunixin; 1 mg/kg, i.m., Napzin^®^, PISA), and a LEV bolus (200 mg/kg, i.p.; UCB Laboratories). At the end of the LEV treatment, the mini-pumps were removed under anesthesia [58,125].

### 4.5. Tissue Sample Collection and RNA Isolation

After treatment, the rats were sacrificed using an anesthetic overdose of pentobarbital (65 mg/kg i.p.). After decapitation, DGs from both hippocampi were micro-dissected [128] and stored in RNase inhibitor solution (RNA Later^®^, Invitrogen, San Diego, CA, USA, EE. UU.) at −80 °C until use. Total RNA was extracted with the RNeasy^®^ Mini Kit (QIAGEN^®^, Hilden, Germany) according to the manufacturer’s instructions. The hippocampal DG (30 mg) was lysed and homogenized using a homogenizer (Omni International TH, TH-01; Kennesaw, GA, USA). The RNA integrity, concentration, and quality were assessed using the QIAxcel advanced equipment (QIAGEN^®^, Hilden, Germany) and a UV–Vis NanoDrop One spectrophotometer (Thermo Fisher Scientific, Waltham, MA, USA). All the extracted RNA was preserved at −80 °C.

### 4.6. Microarray Processing and Data Analysis

The whole transcriptome was assessed using the Affymetrix Rat Gene 2.0 ST microarray, which was designed specifically for the *Rattus norvegicus* genome and provides comprehensive coverage of the transcriptome. The data were processed according to the manufacturer’s instructions (Affymetrix, Santa Clara, CA, USA). The microarray data analysis was performed using the .CEL file of each microarray, grouped by condition (CTRL, EPI, and EPI + LEV), and using the TAC 4.0.2 software (Thermo Fisher Scientific, Waltham, MA, USA). The analysis was performed according to the manufacturer’s instructions and following the principles of the lima package in Bioconductor R. The rn5 (*Rattus norvegicus*) genome was selected, and the microarrays were subjected to inter-microarray normalization for each group and background correction. To determine the differences in gene expression between the groups, the following comparisons were made: EPI vs. CTRL and EPI + LEV vs. EPI groups. A gene was considered differentially expressed when it had a *p* value < 0.05 according to the empirical Bayes method and a false discovery rate (FDR) ≤ 0.05 according to multiple Benjamini–Hochberg corrections. Finally, an FC filter criterion was selected with a cutoff of an FC ≤ −2 and ≥2.

Differentially expressed genes were classified according to the GO system into three broad categories, MF, CC, and BP, using the Database for Annotation, Visualization and Integrated Discovery (DAVID). Fisher’s exact test was used to obtain the *p* values, which were filtered with a cutoff value of 0.05 for a term’s enrichment score. In this study, all annotations were examined, including those with E. scores ≤ 1.3, to ensure that no ontological terms related to pathology were overlooked.

### 4.7. Real-Time Quantitative PCR Analysis

To confirm the differential expression of the genes of interest found in the microarray analysis, a reverse transcription (RT)–PCR technique was used. One-step RT–PCR was performed using the QuantiNova SYBR Green RT–PCR Master Mix Kit (QIAGEN^®^, Hilden, Germany) according to the manufacturer’s protocol and using a Rotor-Gene Q (QIAGEN^®^, Hilden, Germany). The design of the specific primers was based on existing National Center for Biotechnology Information (NCBI) cDNA sequences for *Rattus norvegicus* and was performed using the Primer-BLAST web tool of NCBI [129]. The primer sequences are listed below: *Hgf* F: 5′-CCATGATCCCCCATGAACACA-3′; *Hgf* R: TAGCTTTCACCGTTGCAGGT; *Slc6a13* F: 5′-CACAAGCGCATCCGGTAGA-3′; *Slc6a13* R: 5′-TCCCGAGACCCTGTTATCCA-3′; *Slc24a1* F: 5′-CTTTCTCCTGCCCATCGTGT-3´; *Slc24a1* R: 5′-TATGTTACTGCCCACCGAGC-3′; *Serpinb1a* F: 5′-GGCTGATCTGTCTGGCATGT-3′; *Serpinb1a* R: 5′-TGCAGAATGTAGCGATGCCT-3′; *Adam8* F: 5′-CCCCAAGAC-CTATAGTGAAAACCAA-3′; *Adam8* R: 5′-CAAAGGTTGGCTTGACCTGCTT-3′; Tpt1 F: 5′-GCAGAGCAAATTAAGCACATCC-3′; *Tpt1* R: 5′-ATGGTAAGCAGCAGCCAAT-3′; *Efcab1* F: 5′-GAGATGTGCCAGAGAAGCCA-3′; *Efcab1* R: 5′-TGAACCCACTCTGA-TACGCT-3′; *Lep* F: 5′-AGACCCCAGCGAGGAAAATG-3′; *Lep* R: 5′-TAC-CGACTGCGTGTGTGAAA-3´; *Trhr* F: 5′-GCCACTGTGCTTTACGGGTTTA-3′; *Trhr* R: 5′-CCACTGCAAGCATCTTGGTGA-3´; *B2m* F: 5′-GTGTCTCAGTTCCACCCACC-3′; and *B2m R*: 5′-TTACATGTCTCGGTCCCAGG-3′.

The parameters for reverse transcription were as follows: An amount of 24.5 ng of total RNA was transcribed to complementary deoxyribonucleic acid (cDNA) at 50 °C for 10 min using the RapidScript Hot-Start RT enzyme. Then, the initial activation of DNA polymerase took place for 2 min at 95 °C and was set up for two-stage cycling with a total of 40 cycles, with a denaturation stage at 95 °C for 5 s and an alignment and extension stage at 60 °C for 10 s with fluorescence detection in the alignment and extension stage. A melting curve was generated after amplification by maintaining the temperature at 60 °C for a period of 15 s, followed by a gradual increase (+1 °C) in the temperature up to 95 °C to distinguish nonspecific PCR products.

The comparative delta–delta cycle threshold (^∆∆^CT) method was used to measure the relative expression of the target genes, which was normalized to the housekeeping gene β-2-microglobulin (B2M) [130]. No changes in the B2M transcript levels were observed among the groups under our experimental conditions. The linear portion of the logarithmic amplification curve was used to establish the CT, which was the same CT for the target genes and reference gene. All RT–qPCR plates were performed in triplicate, and curves with a TC < 30 cycles were considered for the calculations.

### 4.8. Statistical Analysis

All the statistical analyses were performed using GraphPad Prism version 9.2.0 software (Boston, USA). The data are expressed as the mean ± standard error of the mean (S.E.M.). The Kolmogorov–Smirnov normality test was performed based on the null hypothesis that the data were normally distributed. For the Gene Ontology (GO) enrichment score data, the results were analyzed using Fisher’s exact test. The RT–qPCR data were analyzed using Student’s *t* test with Welch´s correction or one-way analysis of variance (ANOVA) with Tukey’s post hoc analysis. The SRS data were analyzed using the Mann–Whitney non-parametric rank sum test. The level of significance was set at *p* < 0.05.

## Figures and Tables

**Figure 1 ijms-25-01690-f001:**
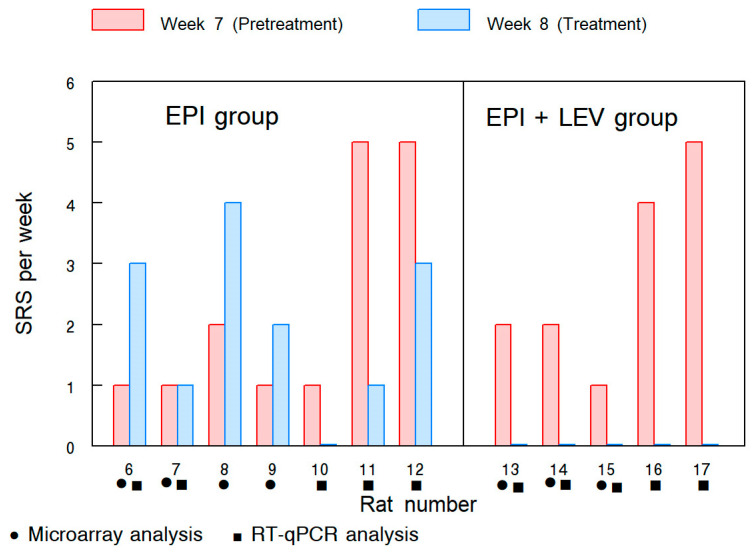
Spontaneous recurrent seizures (SRSs) in epileptic (EPI) and epileptic levetiracetam-treated (EPI + LEV) rats. The data represent the number of SRSs in each animal at week 7 (before treatment) and week 8 (during treatment). • Animals used for microarray dentate gyrus analysis; ▪ animals used for RT–qPCR dentate gyrus analysis.

**Figure 2 ijms-25-01690-f002:**
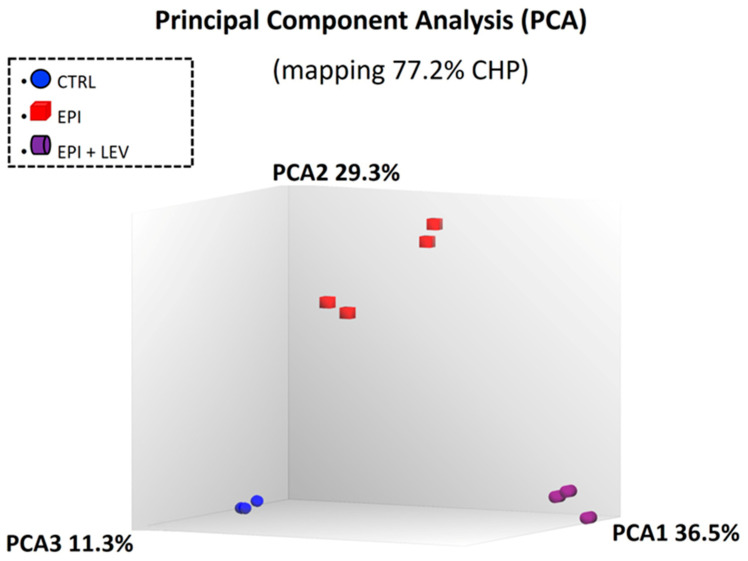
Principal component analysis (PCA) of the microarrays on the dentate gyrus in the control (CTRL), epileptic (EPI), and EPI + LEV (levetiracetam) groups.

**Figure 3 ijms-25-01690-f003:**
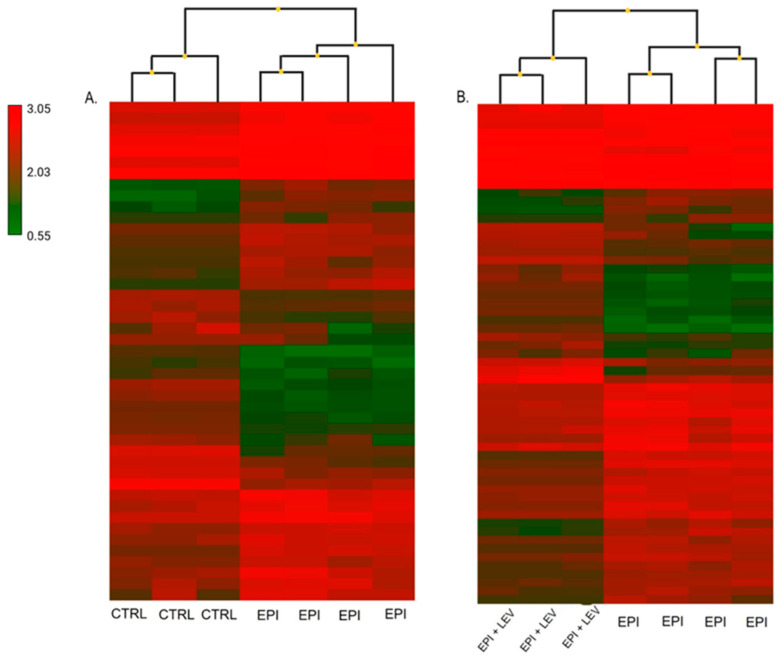
(**A**) Hierarchical cluster analysis of gene expression microarray data from the dentate gyrus (DG) of the control group (CTRL) and the epileptic (EPI) groups. (**B**) Hierarchical cluster analysis of DG microarrays in the groups: EPI and epileptic treated with levetiracetam (EPI + LEV). Red indicates higher and green indicates lower gene expression levels. Each column represents an individual, and each line represents a gene.

**Figure 4 ijms-25-01690-f004:**
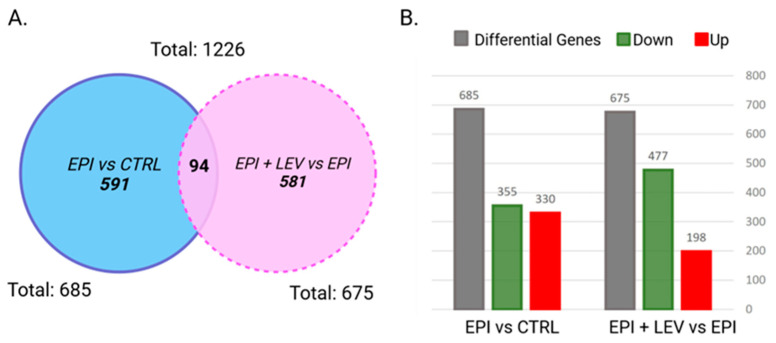
Number of genes showing altered expression in the dentate gyrus of epileptic rats (EPI) and epileptic rats treated with levetiracetam (EPI + LEV). (**A**) Venn diagram of genes detected in each comparison. (**B**) EPI vs. CTRL and EPI + LEV vs. EPI comparisons. Red: upregulated genes; green: downregulated genes. The ANOVA method: ebayes with Benjamini and Hochberg post hoc test, *p* < 0.05; fold change was considered ≤−2 or ≥2.

**Figure 5 ijms-25-01690-f005:**
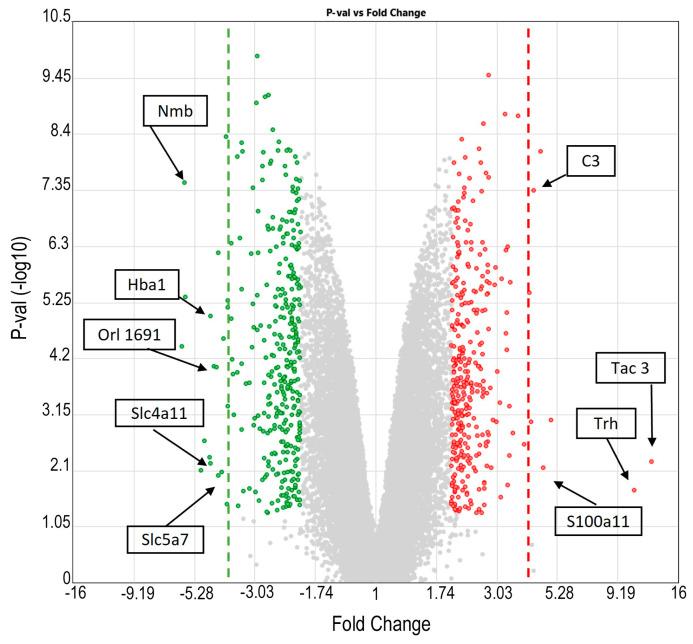
Volcano plot of genes significantly modified by epilepsy (EPI vs. CTRL analysis) with a fold change (FC) ≤−2 or ≥2 and *p*-value > 1.3 (*p* values are expressed as −log10, where a *p*-value > 1.3 is equivalent to *p* < 0.05). The green points are downregulated genes; the red points are upregulated genes. The green dotted line shows an FC ≤ −4, and the red dotted line shows an FC ≥ 4. Complement component 3 (*C3*). Thyrotropin-releasing hormone (*Trh*). Tachykinin 3 (*Tac3*). S100 calcium-binding protein A11 (*S100a11*). Solute carrier family 5 (choline transporter) member 7 (*Slc5a7*). Solute carrier family 4, sodium borate transporter, member 11 (*Slc4a11*). Hemoglobin, alpha 1 (*Hba1*). Olfactory receptor 1691 (*Olr1691*). Neuromedin B (*Nmb*). Unlabeled points with FC ≤ −4 or FC ≥ 4 are genes with unknown official names in the NCBI database.

**Figure 6 ijms-25-01690-f006:**
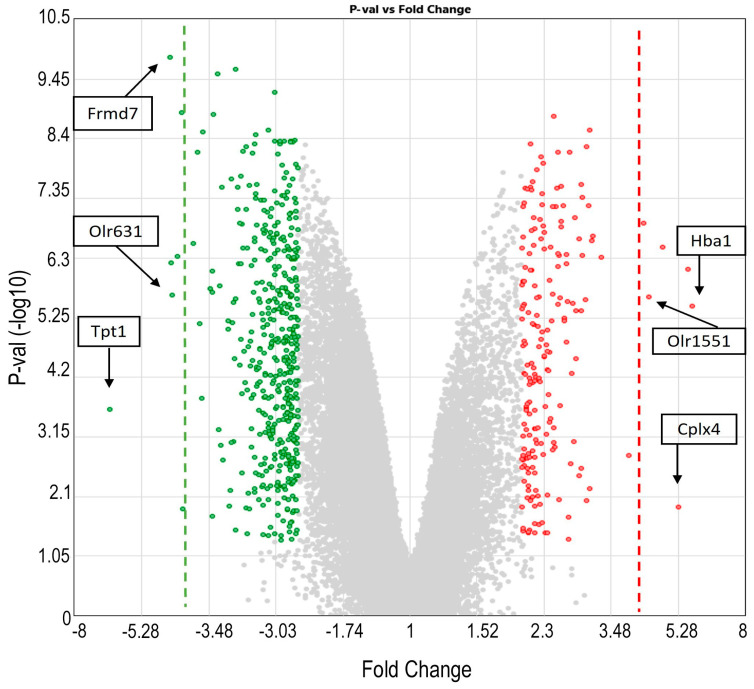
Volcano plot of genes significantly modified by levetiracetam (LEV) treatment in rats with epilepsy (EPI + LEV vs. EPI analysis) with a fold change (FC) ≤ −2 or ≥2 and *p*-value > 1.3 (*p* values are expressed as −log10, where a *p*-value > 1.3 is equivalent to *p* < 0.05). The green points are downregulated genes; the red points are upregulated genes. The green dotted line shows an FC ≤ −4, and the red dotted line shows an FC ≥ 4. Complexin 4 (*Cplx4*). Hemoglobin, alpha 1 (*Hba1*). Olfactory receptor 1551 (*Olr1551*). FERM domain-containing 7 (*Frmd7*). Olfactory receptor 631 (*Olr631*). Tumor protein translationally controlled 1 (*Tpt1*). Unlabeled points with FC ≤ −4 or FC ≥ 4 are genes with unknown official names in the NCBI database.

**Figure 7 ijms-25-01690-f007:**
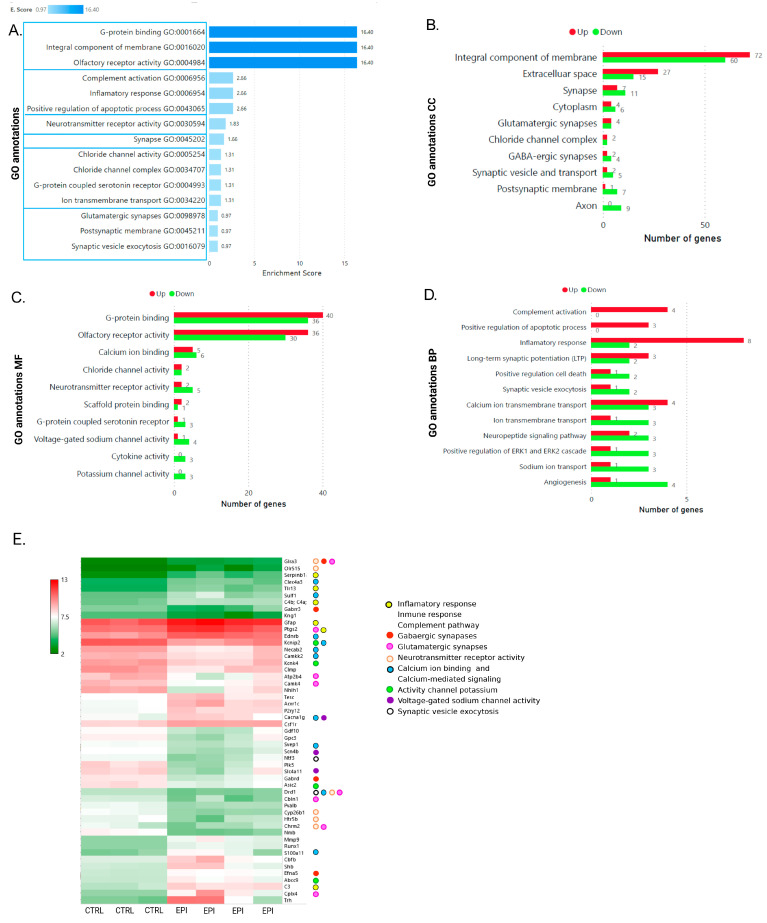
Gene Ontology (GO) analysis of epileptic rats. (**A**) Top enrichment score for all GO terms on differential expression of genes in the comparison of EPI vs. CTRL group; each cluster has the same E. score and is separated from the others by blue lines. GO annotations on (**B**) cellular component (CC); (**C**) molecular function (MF); and (**D**) biological process (BP). (**B**–**D**) The red bars show upregulated genes, and the green bars show downregulated genes. (**E**) Expression levels of genes associated with the following GO terms: inflammatory response (yellow), GABAergic synapses (red), glutamatergic synapses (pink), neurotransmitter receptor activity (light pink), calcium ion binding (blue), activity channel potassium (green), voltage-gated sodium channel activity (purple), and synaptic vesicle exocytosis (white). Fisher’s exact test (enrichment score), *p* < 0.05; background, *Rattus norvegicus*.

**Figure 8 ijms-25-01690-f008:**
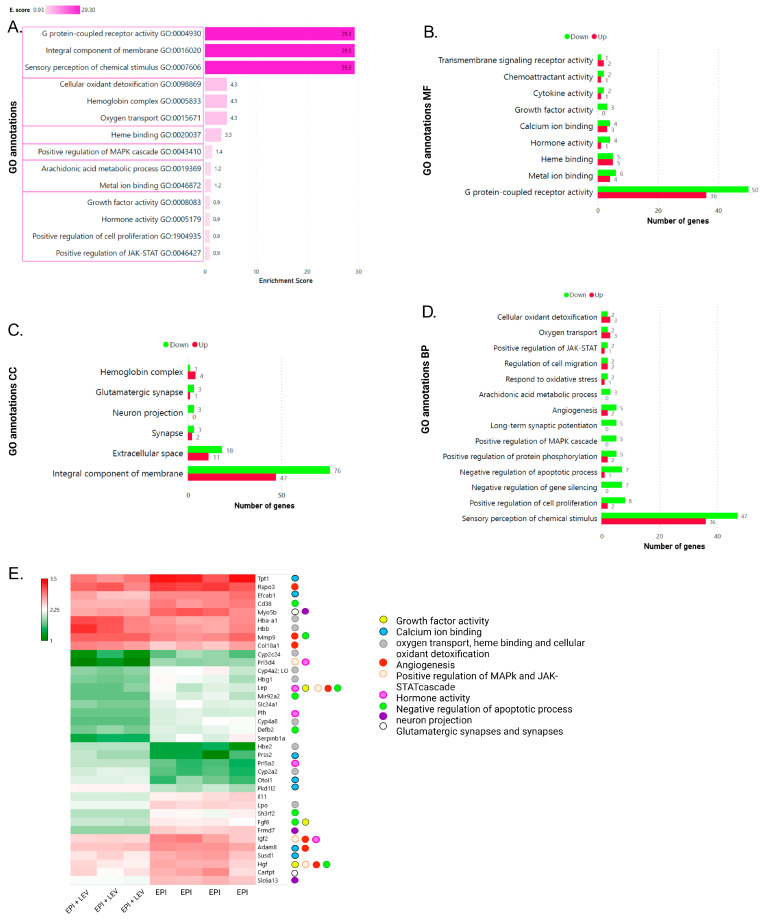
Gene Ontology (GO) analysis of the EPI + LEV vs. EPI comparison. (**A**) Top enrichment score for all GO terms on differential expression of genes in the comparison of EPI + LEV vs. EPI groups; each cluster has the same E. score and is separated from the others by pink lines. (**B**) molecular function (MF); (**C**) Cellular component (CC), and (**D**) biological process (BP). (**B**–**D**) The red bars show upregulated genes, and the green bars show downregulated genes. (**E**) Expression levels of genes associated with the following GO terms were associated with growth factor activity (yellow), calcium ion binding (blue), oxygen transport and heme binding (gray), angiogenesis (red), positive regulation of the MAPK pathway (light pink), hormone activity (pink), negative regulation of the apoptotic process (green), neuron projection (purple), and glutamatergic synapses (white). The gene selection process shown in E was guided by the biological relevance, uniqueness, and interactions of the genes observed in the ontological analysis of the EPI vs. CTRL groups. Fisher’s exact test (enrichment score), *p* < 0.05; background, *Rattus norvegicus*.

**Figure 9 ijms-25-01690-f009:**
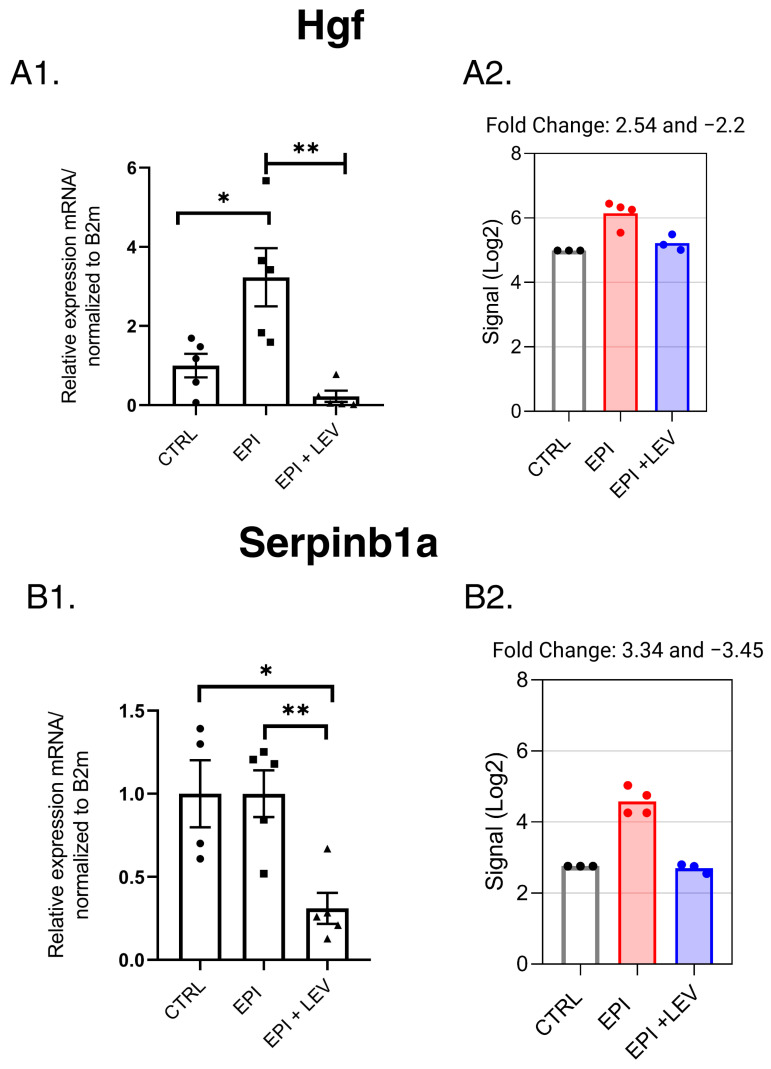
Expression profile of hepatocyte growth factor (*Hgf*) (**A**) and serine (or cysteine) peptidase inhibitor, clade B, member 1a (*Serpinb1a*) (**B**) genes. The expression values obtained from q-PCR experiments are shown in (**A1**,**B1**). The expression values obtained from the gene microarray experiments are shown in (**A2**,**B2**). The data are presented as means ± S.E.M.s; n = 5 per group. * *p* < 0.05, ** *p* < 0.01. One-way ANOVA, post hoc Tukey’s test.

**Figure 10 ijms-25-01690-f010:**
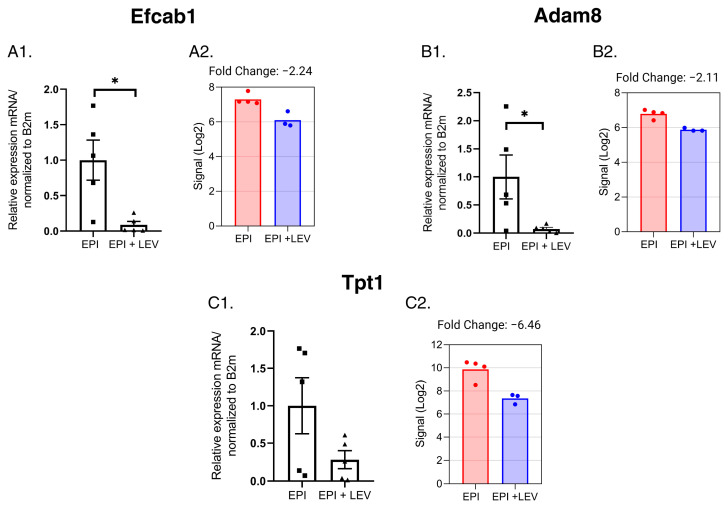
Expression profile of the EF-hand calcium-binding domain 1 (*Efcab1*), ADAM metallopeptidase domain 8 (*Adam8*), and tumor protein translationally controlled 1 (*Tpt1*) genes, modified by LEV and associated with calcium binding and Ca^2+^ homeostasis. The expression values obtained from q-PCR experiments are shown in (**A1**–**C1**). The expression values obtained from the gene microarray experiments are shown in (**A2**–**C2**); n = 5/group. * *p* < 0.05, Student’s *t*-test with Welch’s correction.

**Figure 11 ijms-25-01690-f011:**
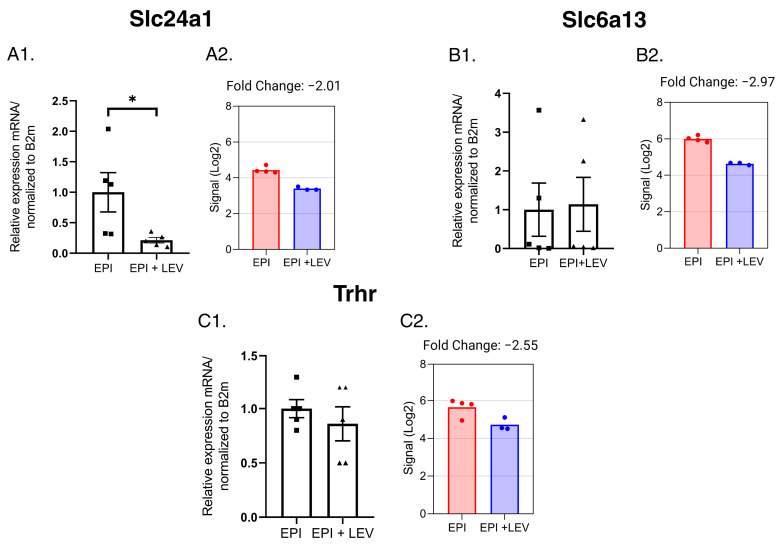
Expression profile of solute carrier family 6 member 13 (*Slc6a13*), solute carrier family 24, sodium/potassium/calcium exchanger, member 1 (*Slc24a1*), and thyrotropin-releasing hormone receptor (*Trhr*) genes, modified by LEV. The expression values obtained from q-PCR experiments are shown in (**A1**–**C1**). The expression values obtained from the gene microarray experiments are shown in (**A2**–**C2**); n = 5/group. * *p* < 0.05, Student’s *t*-test with Welch´s correction.

**Figure 12 ijms-25-01690-f012:**
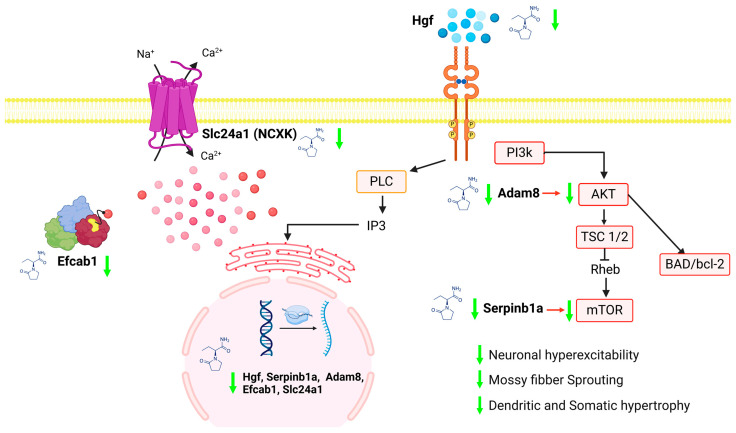
The response to LEV treatment may involve multiple mechanisms. In the epileptic brain, the PI3k/AKT/mTOR pathway (red boxes) is hyperactivated and leads to neuronal hyperexcitability, sprouting of mossy fibers, and dendritic and somatic hypertrophy in parallel with dysregulation and an increase in intracellular Ca^2+^ (red dots). LEV administration in the epileptic brain may attenuate these mechanisms through the downregulation of genes such as hepatocyte growth factor (*Hgf*), ADAM metallopeptidase domain 8 (*Adam8*), and serine (or cysteine) peptidase inhibitor, clade B, member 1a (*Serpinb1a*), which are involved in the PI3k/AKT/mTOR pathway. Moreover, *Hgf*, solute carrier family 24, sodium/potassium/calcium exchanger, member 1 (*Slc24a1*), and EF-hand calcium-binding domain 1 (*Efcab1*participate in the regulation of intracellular Ca^2+^ and subsequently decrease epileptic seizures.

**Table 1 ijms-25-01690-t001:** Effect of levetiracetam (LEV) treatment on the spontaneous recurrent seizure (SRS) number in epileptic (EPI) and LEV-treated epileptic (EPI + LEV) rats.

SRS per Week		
Group	(Pretreatment week)	(Treatment week)
EPI	**1.0** (1.00–4.25)	**2.0** (1.00–3.00)
EPI + LEV	**2.0** (1.75–4.25)	**0.0** (0.00–0.00) *^,†^

The data (medians are listed in bold, and 25th–75th percentiles are listed in parentheses) are presented as the medians of the number of SRSs. * *p* < 0.05 vs. the EPI group; ^†^
*p* < 0.05 vs. the pretreatment week in the same group; n = 5–7 per group (Mann–Whitney rank sum test).

**Table 2 ijms-25-01690-t002:** Genes whose expression was altered by epilepsy or by levetiracetam treatment (intersection of Venn diagram; upregulated genes are highlighted in red and downregulated genes are in green).

GO Term	Gene	Gene Name	FC EPI	FC LEV	*p*-Value
Synapse	*Cplx4*	complexin 4	4.15	5.29	0.0069
Extracellular space; cytoplasm	*Csta*	cystatin A (stefin A)	−3.4	2.58	0.0000119
Increased cell proliferation; angiogenesis; negative regulation of the apoptotic process; regulation of cell migration; chemoattractant activities	*Hgf*	hepatocyte growth factor	2.54	−2.2	0.0029
Angiogenesis; oxidative stress; negative regulation of apoptotic process; regulation of cell migration	*Mmp9*	matrix metallopeptidase 9	2.8	2.14	0.0005
G protein-coupled receptor activity	*Mrgprx2*	Mas-related GPR, member X2	2.15	−2.02	0.0012
Inflammatory response; extracellular space	*Serpinb1a*	serine (or cysteine) proteinase inhibitor, clade B, member 1a	3.44	−3.45	0.0002
Inflammatory response; extracellular space	*SERPING1*	serpin peptidase inhibitor, clade G (C1 inhibitor), member 1	2.67	2.16	0.0027
Nucleus	*Rhox9*	reproductive homeobox 9	−2.04	2.09	0.0004
Nucleus	*Zfp53*	Zinc finger protein 53	2.77	−2.99	0.009
Integral component of plasma membrane; olfactory receptors; sensory perception of chemical stimulus; synapses	* Olr24 *	olfactory receptor 24	−2.63	3.05	0.000178
	* Olr260 *	olfactory receptor 260	2.07	−2.21	0.0141
	* Olr443 *	olfactory receptor 443	2.42	−2.84	0.0027
	* Olr576 *	olfactory receptor 576	−2.11	2.19	0.0022
	* Olr790 *	olfactory receptor 790	2.96	−2.46	0.0002
	* Olr1232 *	olfactory receptor 1232	2.0	−2.07	0.025
	* Olr1251 *	olfactory receptor 1251	−2.08	2.15	0.006
	* Olr1308 *	olfactory receptor 1308	2.8	−3.62	0.000178
	* Olr1456 *	olfactory receptor 1456	−2.54	2.25	0.0006
	* Olr1511 *	olfactory receptor 1511	4.08	−2.56	0.0002
	* Olr1532 *	olfactory receptor 1532	2.8	−2.06	0.01
	* Olr1686 *	olfactory receptor 1686	−2.33	2.17	0.0008

GO = Gene Ontology. FC = fold change. EPI = epileptic group. LEV = levetiracetam-treated epileptic group.

**Table 3 ijms-25-01690-t003:** Groups of animals and the analyses carried out on their brains.

Rat	Group	Microarray Analysis	qPCR Analysis
1	CTRL	X	X
2	CTRL	X	X
3	CTRL	X	X
4	CTRL		X
5	CTRL		X
6	EPI	X	X
7	EPI	X	X
8	EPI	X	
9	EPI	X	
10	EPI		X
11	EPI		X
12	EPI		X
13	EPI + LEV	X	X
14	EPI + LEV	X	X
15	EPI + LEV	X	X
16	EPI + LEV		X
17	EPI + LEV		X

## Data Availability

The data presented in this study are available upon request from the corresponding author.

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
