# Peer review of "Changes in the Dentate Gyrus Gene Expression Profile Induced by Levetiracetam Treatment in Rats with Mesial Temporal Lobe Epilepsy"

_ijms, 2024, doi:10.3390/ijms25031690_

Round 1

Reviewer 1 Report

Comments and Suggestions for Authors

The manuscript is aimed at identifying the molecular genetic mechanisms of the development of pilocarpine-induced Temporal lobe epilepsy (TLE) and the effect of levetiracetam on transcriptome changes in the dentate gyrus of epileptic male Wistar rats. This study is certainly aimed at solving current problems related to the mechanisms of manifestation of socially significant pathology. However, the manuscript contains many problems that need to be corrected before publication.

Major concerns:

Judging by the data in Table 1, in the EPI (treatment week) group, not all animals showed seizures, i.e. they cannot be considered epileptic. This means that the manifestation of epilepsy at this time point is weakened by itself, i.e. without administration of levetiracetam. This needs to be discussed.

137-139: the authors describe genes with maximum differences in transcription levels, but the pvals of most of these genes are close to the critical significance value. This suggests that the scatter in the measurements was very large and the differential expression levels of these genes should be carefully tested using a larger sample. This needs to be discussed.

197: “annotations related to epilepsy” - it is not clear which of the terms presented in Figure 6 are classified by the authors as terms associated with epilepsy, and which do not belong to this list.

235-237: “comparison of CTRL vs EPI group.” “B-D, the red bars show upregulated genes, and the green bars show downregulated genes.” – so what did you analyze EPI vs CTRL or CTRL vs EPI? Understanding which genes were activated in the experiment and which reduced the level of transcription during the development of pathology depends on this. It needs to be clearly stated so that the reader does not get confused.

254-263: in the Venn diagram (Figure 3A) it is shown that the authors made comparisons of EPI vs CTRL and EPI vs EPI + LEV. If this is so, then the inverted sign of FC does not mean restoration, but aggravation of the process of changing expression. Accordingly, 18 genes that are presented in Table 2 were not restored, but continued to change the level of transcription in the same direction as during the induction of epilepsy. The authors should look into their experiment more carefully and correct errors.

413: Serpinb1a – its activation was not confirmed in qPCR; accordingly, the authors cannot discuss the activation of the mTOR pathway with the participation of Serpinb1a

532: The Animals section does not indicate how many rats were taken into the experiment and how many of them were used for transcriptomic analysis and how many for qPCR analysis. Were these rats the same, or were different analyzes performed on different groups of rats?

575: according to the text presented in the Levetiracetam Treatment section, it should be understood that the operation to implant osmotic mini-pumps was carried out only for one group of animals (EPI + LEV). It is necessary to explain why there was no control group with sham-operated rats and why the authors believe that the reduction in the number of seizures is a result of the administration of Levetiracetam, and not the removal of inflammatory processes as a result of the administration of antibiotics during surgery.

 615: CTRL vs EPI groups. – in the results section, two different options are given for what was calculated relative to what. It is necessary to do it equally correctly throughout the text.

654: to carry out qPCR, the authors used the housekeeping gene β-2-microglobulin (B2M). However, in the cited work, the stability of the expression of this gene was related to a dissimilar experiment. The authors must show in the manuscript that, under the conditions of their experiment, this gene does not change the level of transcription.

The authors did not provide complete lists of the found DEGs, which significantly reduces the value of the presented results.

The article does not have a Limitations section. The authors should note that the study was conducted on a limited number of animals (3-4 rats), the experiment is characterized by a large scatter of measurements and, of course, requires additional research before any of the identified genes can be considered candidate genes that have a significant effect on development of an epileptic condition.

Major concerns:

Title: The title is worded as if the manuscript is the first to show that levetiracetam can modulate the manifestation of epilepsy.

28: In the EPI analysis - for those who read the abstract it will remain unclear what was compared with what

30: the EPI+LEV group - for those who read the abstract it will remain unclear what was compared with what

32: serpingb1a - the first letter must be capitalized, in addition, it is not clear which gene the authors mean here (Serpinb1a or Serping1).

35: Serpinb1 – perhaps the authors wanted to write Serpinb1a

Table 1: Animal characteristics are best shown graphically, showing the number of seizures for each animal. The table shows data that there were 5 animals in each group. However, further in the transcriptomic analysis there were fewer animals. Therefore, in Table 1 (better in the figure) it is necessary to show the characteristics of exactly those animals that were then used in transcriptomic analysis.

Figure 1. The choice of shape to display a group of EPI + LEV rats is poor - each dot looks like a double circle.

Figure 2. It is necessary to make a colored legend and show the clustering of samples.

Figure 3b. It is necessary to correct the misspelling of the word: diferencial

Figure 4. It is not clear why the authors made fractional scales of values ​​on the X and Y axes. It is necessary to format the values on the X and Y axes.

149: EPI analysis – it’s not clear why the authors introduce this term, it confuses the reader. It’s better to write what was compared with what (EPI vs CTRL)

150: -log10 (p< 0.05, P-val > 1.3) – perhaps the authors wanted to write -log10 (Pvalue) > 1.3, i.e. p<0.05?

153: solute – after the period should be written with a capital letter

Figure 5. It is not clear why the authors made fractional scales of values ​​on the X and Y axes. It is necessary to format the values on the X and Y axes.

158: E-LEV analysis – it is not clear why the authors introduce this term, it confuses the reader. It’s better to write what was compared with what (EPI vs EPI + LEV)

158: -log10 (p< 0.05, P-val > 1.3) – perhaps the authors wanted to write -log10 (Pvalue) > 1.3, i.e. p<0.05?

In Figures 4 and 5, not all points with FC>|4| are labeled, why? How was the choice made about what to sign and what not to sign?

177: altered terms were found - an incorrect expression, since it is not the terms that have changed, but the terms are enriched with genes that have changed.

182: 72 and 60; 40 and 36; 36 and 30, respectively – an unclear phrase. You need to at least somehow indicate what you mean, for example: 72 up and 60 down; 40 up and 36 down; 36 up and 30 downregulated DEGs, respectively

187, 188 and several other lines: subexpressed – it is customary to use the term downregulated genes

188: genes belonging to this category were in balance - a strange phrase. The word balance is inappropriate here. It is better to rephrase and simply write that the number of DEGs with increased and decreased expression was almost equal.

194-195: Considering that the same gene can belong to different GO processes, we performed a frequency analysis by gene (Figure 6E); - Figure 6E shows only 51 genes; it is not clear on what basis the full list of genes annotated in GO was reduced.

203: Figure 7A shows 6 of these clusters. – it is not clear why 8 clusters were found, but only 6 of them are shown.

In Figures 6 and 7, no boundaries between clusters are visible.Figure 7E: It is not clear on what basis the full list of genes annotated in GO was reduced.

269-272: The text is not clear. The authors should explain what they meant. Does anyone really have goals other than validating the expression of the most likely candidate genes?

276: gene expression patterns were consistent and reproducible (Figures 8, 9 and 10). – this phrase raises questions, since according to the data presented in Figures 8-10, the results of q-PCR did not always correspond to the results of transcriptomic analysis (Serpinb1a, Slc6a13, Tpt1, Trhr), which the authors themselves write about a little below.

279: here it is probably necessary to indicate in the text that further text is already related to qPCR. For example, in qPCR, for Hgf, the relative expression of messenger. This will make it much easier for the reader to understand what you are talking about.

296: In validated genes involved in calcium ion binding – the word validated is used unsuccessfully, because if they are already validated, then why confirm again.

424: serpinb1a – please correct to Serpinb1a

Figure 11: in the figure the authors wrote IP3k, and in the legend PI3k

485-487: the text does not match the picture. From Figure 11, it is most likely that reduction of Serpinb1a and downregulation of mTOR lead to neuronal hyperexcitability, sprouting of mossy fibers and dendritic and somatic hypertrophy. It is necessary to make the drawing more clearly readable.

490: Serpinb1 – perhaps the authors wanted to write Serpinb1a

Comments on the Quality of English Language

187, 188 and several other lines: subexpressed – it is customary to use the term downregulated genes

Reviewer 2 Report

Comments and Suggestions for Authors

The article titled "Levetiracetam Treatment Modifies Alterations in Dentate Gyrus Gene Expression in the Pilocarpine Model of Mesial Temporal Lobe Epilepsy" investigates the effects of levetiracetam (LEV) on gene expression in the dentate gyrus (DG) of epileptic rats. The study uses whole transcriptome microarrays to analyze the differential gene expression in epileptic rats (EPI) and EPI rats treated with LEV for one week (EPI+LEV). The authors also perform quantitative RT-qPCR to confirm the differential expression of genes of interest. 

The manuscript is well-structured and presents a clear and detailed description of the materials and methods used in the study. The authors have employed appropriate statistical analyses to identify differentially expressed genes and have provided comprehensive tables and figures to support their findings. 

However, there are some minor issues with the manuscript: 

1. The introduction could benefit from a more detailed explanation of the role of LEV in the treatment of epilepsy and its potential mechanisms of action. 

2. The authors should provide more information on the selection of genes for validation using RT-qPCR, as the current justification for the choice seems to be based on the authors' expertise. 

3. The discussion could be expanded to provide more interpretation of the results and their implications for the understanding of LEV's mechanism of action in the context of TLE. 

Overall, the manuscript presents a well-conducted and well-analyzed study that contributes to the understanding of LEV's effects on gene expression in the DG of epileptic rats. With minor revisions, the manuscript could be suitable for publication in the journal.

Round 2

Reviewer 1 Report

Comments and Suggestions for Authors

The authors successfully responded to the main comments and made most of the necessary changes to the text of the manuscript. However, the reviewer had several new comments related to changes in the editing of the text. In addition, the authors answered several questions satisfactorily to the reviewer, but did not make the necessary comments in the text of the manuscript. Listed below are necessary additional changes that must be made to the manuscript before publication.

New comments:

Line 33:  LEV treatment reversed the increased expression of Hgf and Serpinb1a mRNA – According to the results of RT‒qPCR analysis, this conclusion is valid only for Hgf. Since the results of analysis of Serpinb1a gene expression using microarrays did not coincide with the results of real-time PCR, the authors cannot claim that LEV treatment reversed the increased expression of Serpinb1a mRNA in their model.

Lines 110-111: Evaluation of the animals' seizure behavior lasted two weeks and began seven weeks after status epilepticus (SE) induction. – It is better to describe events in the correct order, rather than backwards.

Lines 111-113: As shown in Figure 1, at week seven (before treatment), rats in both epileptic groups exhibited SRS with medians (Table 1):

1) Figure 1 does not show weeks. Looking at Figure 1, it is not clear that Pretreatment is actually Pretreatment period (week seven), and treatment should be understood as treatment period (week eight), which makes it very difficult to perceive the information in Figure 1.

2) Figure 1 does not demonstrate SRS with medians. Authors should describe separately the information that the reader sees in Figure 1 and the information that follows from Table 1. From the reviewer's point of view, the version of Figure 1 proposed by the authors is quite confusing for the reader. The confusion occurs due to the fact that, according to Figure 1a, in the eighth week (treatment period), the SRS frequency increases, and according to Figure 1b, in the eighth week (treatment period), the SRS frequency in rats decreases. Perhaps the authors should combine Figures 1a and 1b and point out to readers that the incidence of SRS may either increase or decrease in the EPI group. The added text (lines 116-120) will explain this situation well. In the combined figure, animals selected for microarray analysis and RT‒qPCR analysis should be identified using symbols. After this, of course, readers will have a question about choosing samples for RT‒qPCR analysis. Why did the authors take animal number 10 if it was possible to take samples from animals with a non-zero SRS value in the eighth week? How will the results of RT‒qPCR analysis change if the sample from animal 10 is discarded? From the reviewer's point of view, the sample from Animal 10 is best discarded from the analysis.

Previous comments:

According to the text presented in the Levetiracetam Treatment section (4.4), it should be understood that the operation to implant osmotic mini-pumps was carried out only for one group of animals (EPI + LEV). It is necessary to explain why there was no control group with sham-operated rats and why the authors believe that the reduction in the number of seizures is a result of the administration of Levetiracetam, and not the removal of inflammatory processes as a result of the administration of antibiotics during surgery.

Comments for R1: The authors convincingly answered this question. However, they only did this in responses to the reviewer. It is necessary to include a brief explanation in the text of the manuscript, indicating that the data is not presented in the manuscript.

The authors did not provide complete lists of the found DEGs, which significantly reduces the value of the presented results.

R= As we mentioned in a previous answer, it was not our intention to publish all our data exhaustively but rather to focus on the genetic modifications produced by the administration of LEV to the epileptic brain. Therefore, we decided to simplify the results and present them in a concise way. However, if you consider this necessary, we could add the lists of the DEGs found as supplementary material.

Comments for R1: The reviewer believes that it would be very good manners and respect for readers if the authors present the results of the dynamics of changes in expression in the CNTR, EPI and EPI + LEV groups of at least 94 key genes in the form of supplementary material.

203: Figure 7A shows 6 of these clusters. – it is not clear why 8 clusters were found, but only 6 of them are shown.

R= In Figure 7A, we chose to display only 6 groups because the other 2 groups, which were not included, had an E. score less than 1.3. In accordance with the guidelines of the DAVID bioinformatics tool, we focused on groups with an E score above 1.3. This decision was also made to ensure clear visual presentation, prevent information overload, and direct the reader's attention to the most significant findings. It is important to note that the 6 selected groups are those that demonstrated the greatest biological relevance and a closer relationship with the effects of LEV treatment in epilepsy. Additional details about this selection have been included in sections 2.5 (lines 326-338) and 4.6 of the methodology (lines 726-728).

Comments for R1: The reviewer sees no point in writing about 8 clusters, but presenting 6 that meet the selection criteria. It will be easier and clearer if you immediately write that 6 clusters were found that meet the criteria you selected.

Round 3

Reviewer 1 Report

Comments and Suggestions for Authors

The authors responded to all reviewer comments. The reviewer has 2 more small comments that perhaps the authors should also take into account and make appropriate changes to the text of the manuscript.

Lines 35-36: “and decreased the expression of the Efcab1, Adam8 and Slc24a1 genes in the DG” - the authors can add the Serpinb1a gene here (deleted in the previous sentence)

Lines 128-129: “In particular, one rat in the EPI group did not exhibit SRS during the treatment week;” – instead of “treatment week” it is better to write “week 8”.

Author Response

Thank you for your feedback, we considered the two small comments and made the additional suggested changes to the manuscript:

a) Lines 35-36: “and decreased the expression of the Efcab1, Adam8 and Slc24a1 genes in the DG” – the authors can add the Serpinb1a gene here (deleted in the previous sentence)

RESPONSE: As suggested, Serpinb1a was included in genes downregulated by LEV (lines 35-36).

b) Lines 128-129: “In particular, one rat in the EPI group did not exhibit SRS during the treatment week;” –instead of “treatment week” it is better to write “week 8”.

RESPONSE: We agree, -week 8- is used instead of -treatment week- (lines 118-119).